# Characterization of Two Mitogenomes of *Hyla sanchiangensis* (Anura: Hylidae), with Phylogenetic Relationships and Selection Pressure Analyses of Hylidae

**DOI:** 10.3390/ani13101593

**Published:** 2023-05-10

**Authors:** Yue-Huan Hong, Hai-Ming Huang, Lian Wu, Kenneth B. Storey, Jia-Yong Zhang, Yong-Pu Zhang, Dan-Na Yu

**Affiliations:** 1College of Life Sciences, Zhejiang Normal University, Jinhua 321004, China; 2Zhoushan Forestry Center, Zhoushan 316021, China; 3Department of Biology, Carleton University, Ottawa, ON K1S 5B6, Canada; 4Key Lab of Wildlife Biotechnology, Conservation and Utilization of Zhejiang Province, Zhejiang Normal University, Jinhua 321004, China; 5College of Life and Environmental Science, Wenzhou University, Wenzhou 325035, China

**Keywords:** *Hyla*, Dryophytes, mitogenome, phylogeny, positive selection

## Abstract

**Simple Summary:**

Hylidae is a rich family of Anura that is widely distributed across the world. Previous studies have shown that the *Hyla* and *Dryophytos* genera in the Hylidae used to belong to the same genus. The Eurasian species (including *Hyla sanchiangensis*) originated in North America and spread through the Beringian Land Bridge to Asia during the last ice age. Adaptation to low temperatures may require more energy expenditure. Mitochondria are the center of energy metabolism and, through oxidative phosphorylation, provide most of the ATP energy for the physiological and biochemical activities of the body. The mitogenome was used to investigate whether *Hyla* and *Dryophytos* genera are subject to positive selection. In addition, as a unique species in China, the distinction between *Hyla sanchiangensis* (Anura: Hylidae) and two different sites was also compared and analyzed in this paper.

**Abstract:**

*Hyla sanchiangensis* (Anura: Hylidae) is endemic to China and is distributed across Anhui, Zhejiang, Fujian, Guangdong, Guangxi, Hunan, and Guizhou provinces. The mitogenomes of *H. sanchiangensis* from two different sites (Jinxiu, Guangxi, and Wencheng, Zhejiang) were sequenced. Phylogenetic analyses were conducted, including 38 mitogenomes of Hylidae from the NCBI database, and assessed the phylogenetic relationship of *H. sanchiangensis* within the analyzed dataset. Two mitogenomes of *H. sanchiangensis* showed the typical mitochondrial gene arrangement with 13 protein-coding genes (PCGs), two ribosomal RNA genes (12S rRNA and 16S rRNA), 22 transfer RNA (tRNA) genes, and one non-coding control region (D-loop). The lengths of the 12S rRNA and 16S rRNA genes from both samples (Jinxiu and Wencheng) were 933 bp and 1604 bp, respectively. The genetic distance (*p*-distance transformed into percent) on the basis of the mitogenomes (excluding the control region) of the two samples was calculated as 4.4%. *Hyla sanchiangensis* showed a close phylogenetic relationship with the clade of (*H. annectans* + *H. tsinlingensis*), which was supported by ML and BI analyses. In the branch-site model, five positive selection sites were found in the clade of *Hyla* and *Dryophytes*: Cytb protein (at position 316), ND3 protein (at position 85), and ND5 protein (at position 400) have one site, respectively, and two sites in ND4 protein (at positions 47 and 200). Based on the results, we hypothesized that the positive selection of *Hyla* and *Dryophytes* was due to their experience of cold stress in historical events, but more evidence is needed to support this conclusion.

## 1. Introduction

Amphibians are regarded as useful models for researching the elements that influence patterns of genetic differentiation and diversity. Firstly, they show relatively stable genetic traits as compared to other animals [1]. Secondly, they are very sensitive to climate conditions, and researchers can compare past and present climate changes via changes in various indicators of amphibians [2]. As a result, it is worthwhile to explore the relationship between amphibians and the environment. Hylidae is one of the largest families of frogs, currently comprising over a thousand species from around the world. There are 35 officially named species in the genus *Hyla*, 16 of which are found in Northern and Central America and 19 in Eurasia, including eight species found in China (http://www.amphibiachina.org/, accessed on 26 August 2022) [3]. The genus *Hyla* was shown by Li et al. [4] to have originated in North America and subsequently spread to China via Beringia during the Middle Eocene and Early Oligocene [5]. The genus *Dryophytes* and *Hyla* diverged from other Hylidae about 22.6 million years ago [6]. For both genera, the genus *Hyla* was generally thought to be distributed in Asia, and the genus *Dryophytes* was found mainly in North America [7]. The genera *Dryophytes* and *Hyla* are species of North American origin that were cold adapted during early historical events [4]. Therefore, it is considered to be better adapted to low temperatures than other species of Hylidae, so it may have undergone positive selection.

*Hyla sanchiangensis* [8] (Anura: Hylidae) was first described in 1929 from Wuyi Mountain, Fujian, China. The species is endemic to China and is distributed across Anhui, Zhejiang, Jiangxi, Hubei, Fujian, Guangdong, Guangxi, Hunan, and Guizhou provinces (http://www.amphibiachina.org/, accessed on 26 August 2022) [3]. All these distribution areas are discontinuous in China. These frogs inhabit paddy fields or can be found around fences. As a result of deforestation and habitat degradation, chemical pollution, and disease, *H. sanchiangensis* has declined substantially in Zhejiang province and is listed as a key species of wildlife under protection in Zhejiang province. Through the morphological analysis of *Hyla sanchiangensis* from Zhejiang and Guangxi, it was found that there were some differences in its morphological characteristics. Some studies have indicated that anuran larval development and variation in environmental factors can result in morphological changes that lead to increased intraspecific variation [9,10,11,12,13,14,15]. This means that a single species living in different areas may show significant differences.

Mitogenomes are currently used as an efficient tool for species identification and the analysis of phylogenetic relationships [16,17]. The length of frog mitogenomes is about 15–23 kb [18,19,20,21] and encodes 22 transfer RNAs (tRNAs), two ribosomal RNAs (12S and 16S RNAs), 13 protein-coding genes (PCGs), and the control region (CR) or D-loop region. As a result of their relatively conserved gene content, rapid evolutionary rate, small size, maternal inheritance, and limited recombination, mitogenomes have been widely used for studies of conservation biology, species delimitation, and the evolution of the Anura [16,22,23]. Using the mitogenome to study phylogenetic relationships is an efficient tool to analyze problems in taxonomy. In addition, further studies have found that mtDNA has evolved a variety of unique properties that enable this genome to be used as a cellular sentinel for genotoxic stress. [24]. Based on all of these factors, samples of *H. sanchiangensis* were collected from different sites to conduct a basal study and evaluate the specific differences in the characteristics of *H. sanchiangensis* from two different sites. These data allowed us to further explore the phylogenetic relationship of *H. sanchiangensis* among Hylidae.

Despite numerous studies, the phylogenetic relationships among Hylidae are still controversial, and full confirmation of the monophyly of the genus *Hyla* and the genus *Dryophytes* using mitogenomes still needs full confirmation. *Dryophytes* were long treated as a subgenus of *Hyla* [25], but Duellman et al. [6] suggested that the species belonging to the genus *Hyla* should now be divided into two genera: *Hyla* and *Dryophytes*; subsequent research by Zhang et al. also supported this view [26]. Therefore, it is necessary to further study whether *Dryophytes* and *Hyla* are monophyletic and show the relationships between these two clades. Li et al. [4] supported *H. annectans* and *H. tsinlingensis* as clustering into one clade. However, Lee et al. [27] found that *H. tsinlingensis* and *H. suweoninsis* were sister clades.

Despite the widespread belief that mitogenomes are in a neutral or nearly neutral state of selection [28], there is considerable evidence showing that positive selection acts on mitochondrial genes involved in environmental adaptation, which suggests that mitochondrial genes can be used to explore the relationship between environmental conditions and natural selection [17,29,30]. Mitogenomes of *H. sanchiangensis* from two different sites have been sequenced and compared with the mitogenome of a third site of *H. sanchiangensis* (MT561180), as recently reported from Lishui (LS), Zhejiang [31]. Hence, the current study generated and analyzed the mitogenome of *H. sanchiangensis* and conducted phylogenetic analyses of *Hyla* and *Dryophytes* to explore the relationships among these groups. In addition, positive selection sites in the *Hyla* and *Dryophytes* clades were analyzed to determine whether the clades of *Hyla* and *Dryophytes* were positively selected for cold origin in the mitogenome.

## 2. Materials and Methods

### 2.1. Sample Collection and DNA Extraction

Samples of *H. sanchiangensis* were collected from Jinxiu, Guangxi, China (JX) (24.14° N, 110.18° E) and Wencheng, Zhejiang, China (WC) (27.08° N, 120.08° E). According to the morphology of the frogs, they were preliminarily identified as *H. sanchiangensis* (Anura: Hylidae). Tissue samples were taken via toe clips and preserved at −40 °C in 100% ethanol for subsequent DNA extraction. Total genomic DNA was extracted using an Ezup Column Animal Genomic DNA Purification Kit (Sangon Biotech Company, Shanghai, China), following the manufacturer’s handbook.

### 2.2. Primer Design, PCR Amplification, and Sequencing

PCR amplification of mitogenomes by the polymerase chain reaction (PCR) used universal primers as described by Zhang et al. [32]. Sequence proofreading followed, and specific primers were designed using Primer Premier 5.0 (Primer Biosoft International) based on amplified fragments (Appendix A). The reaction volume for PCR amplification was 50 μL. PCR amplification used standard methods as in Cai et al. [33]. The PCR products were identified by 1% agarose gel electrophoresis, and all PCR-purified products were sequenced at Sangon Biotech Company (Shanghai, China).

### 2.3. Mitogenome Annotation and Sequence Analyses

DNASTAR v.6.0 [34] was used to assemble the results from the two samples and obtain whole mitogenome sequences. MITOS (http://mitos.bioinf.uni-leipzig.de/index.py, accessed on 26 August 2022) [35] was used to determine all tRNA genes. We identified and annotated 12S rRNA, 16S rRNA, and the 13 protein-coding genes (PCGs) by Mega 11.0 [36] and compared the homology of genes [37]. We used tRNAscan-SE1.21 software [38] (http://lowelab.ucsc.edu/tRNAscan-SE/, accessed on 26 August 2022) to predict all the cloverleaf secondary structures of the tRNA genes. In addition, the secondary structure of the light-strand origin region was built by RNAalifold with default parameters (http://rna.tbi.univie.ac.at/cgi-bin/RNAWebSuite/RNAalifold.cgi, accessed on 26 August 2022) [39,40,41]. Mega 11.0 [36] was used to translate the PCG sequences into amino acids according to the vertebrate mitogenome genetic code and codon usage. The AT content and relative synonymous codon usage (RSCU) of the two newly sequenced mitogenomes were also calculated. Finally, AT-skew = (A − T)/(A + T) and GC-skew = (G − C)/(G + C) were used to calculate composition skewness [42].

### 2.4. Genetic Distance

The genetic distance of the two mitogenomes was determined using Mega 11.0 [36] and the *p*-distance approach. To further assess the genetic distance between Hylidae species, we downloaded the RAD-Seq reads from Borzée et al. [43] for three Hylidae species (*Dryophytes suweonensis*, *Dryophytes immaculatus*, and *Dryophytes flaviventris*) as shown in Appendix A and assembled the mitochondrial genes from this whole genome data using GetOrganelle 1.7.1 [44,45,46,47,48].

### 2.5. Phylogenetic Analyses

We downloaded 38 mitogenomes from the NCBI (a matrix of publicly available mitogenomes of the family Hylidae) to investigate the evolutionary connections of the two newly sequenced mitogenomes of *H. sanchiangensis*. Phylogenetic analyses were performed with 40 species from the Hylidae according to the methods of Zhou et al. [49], Yu et al. [50,51], and Zhang et al. [32]. These 40 species are shown with the GenBank accession numbers in Table 1. The mitogenomes of *Dendrobates auratus*, *Mannophryne trinitatis*, *Odontophrynus occidentalis,* and *Rhinoderma darwinii* were used for outgroup rooting [32] (Table 1). Clustal W in Mega 11.0 [36] was used to align the 13 protein-coding genes and translate the gene sequences into amino acids. Conserved regions were selected using Gblock 0.91b [52] with default settings. A dataset of 10,548 nucleotides and 3516 amino acids was obtained after final alignment. The saturation of the first, second, and third codons was analyzed using DAMBE [53]. The third codon positions were not saturated, and the datasets could be used to construct a phylogenetic tree (the results were shown in Appendix A). The optimal partitions and best-fitting model of evolution were selected using PartitionFinder 1.1.1 [46] using the Bayesian information criterion (BIC) [54] (Appendix A). RAxML allows only one rate heterogeneity model in partition analysis. The GTR + I + G model from RAxML 8.2.0 [55] was selected for maximum likelihood (ML) analysis, with each node branch supported for evaluation under 1000 ultra-fast replications. Partitioning results were used via MrBayes version 3.2 [56] for Bayesian inference (BI) analysis and ran four chains for 10 million generations, sampling every 1000 generations. The first 25% of runs were discarded as burn-in according to convergence (<0.01). The rest were used to build the BI phylogenetic tree.

### 2.6. Detecting Selective Pressure

The selection pressure of the mitogenomes was analyzed by the EasyCodeML program [67]. We use *ω* ratio to indicate the natural selection of all 13 PCGs. The *ω* ratio (*dN/dS*) is the rate of nonsynonymous (*dN*) versus synonymous (*dS*) substitution; *ω* = 1 denotes neutral mutations, *ω* < 1 denotes negative selection, and *ω* > 1 denotes positive selection [68].

The genera *Hyla* and *Dryophytes* were chosen as the foreground branches, whereas other frog species excluding outgroups were chosen as the background branches in order to investigate whether the clade of *Hyla* and *Dryophytes* is subject to positive selection. The branch model, branch-site model, and clade model were the three analysis methods utilized to investigate the association between mitochondrial genes and adaptive evolution. In order to obtain the difference between the foreground branch and the background branch, the branch model compared the results obtained by the one-ratio model (M0) and the two-ratio model, respectively. Additionally, the clade model was utilized to examine them in order to analyze numerous clades at once [58]. Finally, to understand the effect of forward selection on some sites in the foreground branch, the A model and the A null model in the branch-site model were tested. These models were also evaluated using LRT, and the posterior probability of positive selection sites was assessed using Bayesian Empirical Bayes (BEB). UniProt was used to collect the structural and functional details of positive selection genes [69]. Furthermore, SWISS-MODEL Workspace was used to construct the three-dimensional structure (3D) of the amino acid positive selection of the corresponding protein [70].

## 3. Results

### 3.1. Mitogenome Organization and Structure

In the present study, two mitogenomes of *H. sanchiangensis* were sequenced from Jinxiu (JX) and Wencheng (WC). The sequences were submitted to GenBank with accession numbers MK388867 and MK388868, respectively. Both are incomplete mitogenomes, missing a portion of the control region. The mitogenomes of *H. sanchiangensis* (WC) were 15,977 bp long, and those of *H. sanchiangensis* (JX) were 15,675 bp long. Both mitogenomes included 13 protein-coding genes (PCGs), two rRNAs (12S and 16S rRNAs), 22 tRNAs, and one control region (partially sequenced) (Figure 1 and Table 2). This arrangement is the same as that of other species of the genus *Hyla*.

The mitogenome compositions of *H. sanchiangensis* from Wencheng and Jinxiu are shown in Table 3, along with the mitogenome of another *H. sanchiangensis* sample reported recently (Genbank #MT561180). The AT% of the nucleotide composition of *H. sanchiangensis* from Wencheng was higher than Jinxiu, and both of them were higher than the *H. sanchiangensis* mitogenome of the individual from Lishui [31] (Table 3). The mitogenome of *H. sanchiangensis* from Lishui exhibited a negative GC-skew (−0.275) and a slightly positive AT-skew (0.001) [31], which was very similar to the genome from Jinxiu.

### 3.2. Protein-Coding Genes and Codon Usages

The length of PCGs from Wencheng and Jinxiu were both 11,331 bp, whereas the length of PCGs from Lishui was 11,316 bp [31]. All PCGs, with the exception of *ND6*, were located on the heavy strand (H-strand). In the mitogenome of *H. sanchiangensis*, the start codons of most genes were ATG, but four genes had modified start codons. Complete stop codons were found in most genes. For three genes (*COX2*, *ND1*, and *ND3*), an incomplete stop codon (T) was identified (Table 2). All of the PCGs had the same start and termination codons in the three mitogenomes of *H. sanchiangensis* analyzed here.

The nucleotide AT% composition of the PCGs for both newly sequenced *H. sanchiangensis* mitogenomes (Wencheng and Jinxiu) was 59.6% and 59.8%, respectively. We calculated the relative synonymous codon usage (RSCU) of the 13 protein-coding genes from the two samples, excluding stop codons. This data is summarized in Figure 2, along with the mitogenome of *H. sanchiangensis* from Lishui. The RSCU showed that codons that had base A or T in the third position were consistently overused as compared to the other synonymous codons. The frequency of amino acid usage was almost the same among the three analyzed mitogenomes (Figure 2). For the mitogenomes of Wencheng and Jinxiu, Leu (CUR), Ala (GCR), and Ile (AUR) (>300) were the most frequently encoded amino acids, and Cys (UGR) (<45) was the least frequently used amino acid. This was similar to the mitogenome of *H. sanchiangensis* from Lishu [31]. Fourteen amino acids had two kinds of codons, whereas Ala, Arg, Leu, Ser, Val, Pro, Thr, and Gly had four kinds of codons. These results indicated that the RSCU of the three mitogenomes was highly conserved.

### 3.3. Ribosomal and Transfer RNAs

The two mitogenomes of *H. sanchiangensis* from Wencheng and Jinxiu contained two rRNAs and 22 tRNA genes. The positions of the 12S rRNA and 16S rRNA were the same as the mitogenome of *H. sanchiangensis* from Lishui [31], and their lengths were highly consistent. In all three sites of *H. sanchiangensis*, it was found that the content of A was higher than that of T, and the content of C was higher than that of G.

A total of 22 tRNAs that had a similar structure were detected in the newly sequenced mitochondrial genomes, and among them, individuals collected at Jinxiu and Wencheng had the same total tRNA length of 1533 bp. Excluding trnA, trnC, trnP, trnQ, trnN, trnY, trnS (UCN), and trnE, all other tRNAs were on the H-strand. The content of A + T was remarkably higher than that of G + C in all three samples from *H. sanchiangensis*.

Based on Appendix A, the secondary structures of the 22 tRNAs were found to be mainly typical clover structures, and all tRNA genes had equivalent lengths except trnK, which was 73 bp and 72 bp in the mitogenomes of the individuals collected at Jinxiu and Wencheng, respectively. In the mitogenome of the individual collected at Wencheng (Appendix A), trnC did not form a dihydrouridine (DHU) loop and trnS (AGN) had lost the DHU arm and could not form a complete cloverleaf secondary structure. In addition, some tRNA genes had unmatched base pairs, such as A-C in trnI, U-U in trnN, and A-A in trnD. In the mitogenome of the individual collected at Jinxiu (Appendix A), the cloverleaf secondary structure was very similar to that of *H. sanchiangensis* from Wencheng. The trnS (AGN), which lost the DHU arm, was a typical feature and had an unusual function. In addition, the trnC also lost the DHU loop. Furthermore, the secondary structure contained non-Watson-Crick base pairs in the structure of stems, such as A-A in trnL (UUN), U-U in trnN, and A-C in trnI.

### 3.4. Intergenic Regions and L-Strand Origin of Replication

There were eight non-coding regions (from 1 to 36 bp) and nine overlapping regions (from 1 to 25 bp) in the mitogenome of *H. sanchiangensis* from Wencheng and Jinxiu (Table 3), as was also described for the mitogenome of *H. sanchiangensis* from Lishui [31]. The longest overlapping region (25 bp) occurred between *ATP8* and *ATP6*. The longest spacer region was 36 bp and occurred between trnS1 and *ND5* in all three mitogenomes.

The most remarkable genomic feature observed in the mitogenomes of *H. sanchiangensis* was the presence of a distinctive insertion between trnN and trnC called the L-strand origin of replication (O_L_), which plays an important role in replication. The O_L_ between trnN and trnC was about thirty bases and showed a similar secondary structure (stem-loop structure) (Figure 3). The structure of the O_L_ was highly similar and symmetrical in the mitogenomes of the individuals collected at Jinxiu and Wencheng, and the mitogenome of the individual collected at Jinxiu only lacked an A in the circle of the stem-loop. The structures of the O_L_ in the mitogenomes of the individuals from Wencheng and Lishui were exactly the same (Figure 3). Furthermore, the replication initiation region of the light strand (O_L_) in the mitogenome of the individual collected in Wencheng was one base longer than that of the mitogenome of the individual collected in Jinxiu. There were nine base pairs in the stem loop, including six pairs of C-G and three pairs of U-A.

### 3.5. Genetic Distance

The amino acid alignment was the backbone to obtain the alignment of 10,983 nucleotide sites, with 5328 variable sites and 4061 parsimonious informative sites. The overall mitogenome genetic distance (excluding the control region) (Appendix A) between individuals of *H. sanchiangensis* collected at Wencheng (MK388867) and Jinxiu (MK388868) was 4.4%, which was the same as the genetic distance observed between the mitogenomes of the individuals of *H. sanchiangensis* collected at Jinxiu (MK388868) and Lishui (MT561180). However, the mitogenome genetic distance between the individual *H. sanchiangensis* collected at Lishui (MT561180) and Wencheng (MK388867) was only 0.2%. On this basis, we concluded that the lower genetic distance between *H. sanchiangensis* collected at Lishui (MT561180) and Wencheng (MK388867) was due to their close geographical locations. Two parts of the *COX1* gene from the known species of Hylaidae were used to calculate genetic distance (Appendix A), and we found that the highest genetic distance between *Dryophytes suweonensis* and *Dryophytes flaviventris* was 1.9%.

### 3.6. Phylogeny of Hylidae

The results of the two phylogenetic analyses (ML and BI) yielded a similar topological structure (Figure 4), and BI was used as the main one. The phylogenetic analyses supported the monophyly of *Boana*, *Hyla,* and *Dryophytes*. Two clades (clades A and B) were formed within Hylaidae. The clade A of (*Nyctimystes* + (*Phyllomedusa* + *Pithecopus*)) and the clade B of ((*Bokermannohyla* + *Boana*) + (*Osteocephalus* + (*Pseudis* + (*Hyla* + *Dryophytes*)))) were well supported. The results showed that *Hyla tsinlingensis* and *H. annectans* have sister-group relationships. *Hyla sanchiangensis* Wencheng (WC) was a sister clade to *H. sanchiangensis* Lishui (LS), and then the clade of (*H. sanchiangensis* WC + *H. sanchiangensis* LS) is a sister clade of *H. sanchiangensis* Mingxi (MX). Finally, this clade got together with *H. sanchiangensis* Jinxiu (JX).

### 3.7. Detecting Selective Pressure

The BI tree, which has the same topology as the ML tree, was used to analyze the selection pressure of the 13 PCGs. In the branch model, there was no role for selection. In this study, *Hyla* and *Dryophytes* were used as foreground clades, respectively, and the rest as background clades for selection pressure analyses. The results showed that in the branch-site model, forward selection occurs only when *Hyla* and *Dryophytes* act as foreground branches. In the branch-site model, model A vs. model A null was significant (*p* < 0.05), with five amino acid selection sites and BEB values >0.95 (amino acid residue 1595 in *Cytb*, 2308 in *ND3*, 2479, and 2632 in *ND4*, 3272 in *ND5*) (Table 4). The distinctive characteristics of these positive selection sites from the *Hyla* and *Dryophytes* clades were investigated to ascertain their functional importance. The results from other models are shown in Appendix A. Table 5 lists the characteristics and descriptions of positive selection sites found in the mitochondrial PCGs of Hylidae species. It can be seen that four positive selection sites were located in the protein transmembrane domain: two were described as Proton_antipo_M, a second as Oxidored_q5_N, and the other as CYTB_CTER. All this evidence suggested that some amino acid sites in the clade of *Hyla* and *Dryophytes* might have been subjected to positive selection.

## 4. Discussion

### 4.1. Mitogenome Structure

The gene arrangements were the same as the mitogenome patterns of other *Hyla* and *Dryophytes* species [26,27,31,59,60,61,62,63,64,65]. Comparing the two mitogenome sequences produced in our study to a previous mitogenome of *H. sanchiangensis* from Lishui (MT561180) [31], subtle differences were found. Some PCGs (e.g., *COX2*, *ND1*, and *ND3*) had an incomplete stop codon, as already identified in other frog species [27,31]. The secondary structure of tRNAs in the two mitogenome sequences contained non-Waston-Crick base pairs in the structure of stems, which was similar to the mitogenome of the individual collected in Lishui [31]. There were many overlapping regions in the three mitogenomes of *H. sanchiangensis*, with the longest overlapping region (25 bp) being between *ATP8* and *ATP6*, which was also found in all mitogenome sequences of *Hyla* species [31,62,63,64,65], whereas only a 10 bp overlap region was found in the *Dryophytes* sequences [26,27,59,60,61]. This further illustrated the differences between the genera *Hyla* and *Dryophytes* and also supported the result expounded by Zhang et al. [26]. In addition, the most remarkable genomic feature was O_L,_ which played an important role in replication when this region was a single chain [71].

### 4.2. Genetic Distance and Phylogeny of Hylidae

According to the results of the current study, the genetic distance between *H. sanchiangensis* individuals collected in Jinxiu and Wencheng or between Jinxiu and Lishui individuals was found to be 4.4%, which was much higher than the genetic distance reported between the species of *Dryophytes* [43]. This suggests that the intraspecific genetic distance of *H. sanchiangensis* was higher than the interspecific distance of *Dryophytes* species. Therefore, it was speculated that cryptic species may exist in *H. sanchiangensis.*

The BI and ML trees constructed from the mitogenomes showed similar topological structures. The clade formed by the individuals of *H. sanchiangensis* collected at Wencheng and Lishui clustered together and formed a sister clade to the *H. sanchiangensis* individual collected at Jinxiu and then formed together with *H. sanchiangensis* collected at Mingxi (Figure 4). This was the same as the results obtained from the mitogenome genetic distance data. The clade of all *H. sanchiangensis* forms the sister clade with (*H. annectans + H. tsinlingensis*), as already identified by Yan et al. [72], while the monophyly of *Hyla* and *Dryophytes* was still well supported in this study. Previously, *Dryophytes* had been treated as a subgenus of *Hyla* [25]. However, our data support the division of these frogs into two clades, *Dryophytes* and *Hyla*, according to their genetic distance and phylogenetic relationships, as well as the results of Huang et al. [73]. The overlap region (25 bp or 10 bp) between *ATP8* and *ATP6* could be used as a molecular characteristic to distinguish the two genera. Due to limited molecular data, our study provided a revised phylogenetic relationship for Hylidae and further promoted the continuing development of mitochondrial genomics.

### 4.3. Detecting Selective Pressure

Positive selection was found when *Hyla* and *Dryophytes* were used as foreground branches in the branch-site model. It was determined that *Hyla* and *Dryophytes* were a single genus in the past, originating from the Eocene to early Oligocene in North America, and the Eurasian species originated in North America and spread through the Beringian Land Bridge [4,5,25,74]. The hypothesis was that the frogs of these two genera were better adapted to low temperatures due to cold adaptation in low temperature regions during early historical events. However, this was just a hypothesis we put forward, and whether it is a fact needs to be proven by further studies.

Four mitochondrial PCGs were positively selected, showing two positive selection sites on *ND4* and one positive selection site on each of the *ND3*, *ND5*, and *Cytb* genes. In addition, according to our branch-site model analysis, ND3, ND4, and ND5 proteins belonged to mitochondrial complex I, and Cytb proteins belonged to mitochondrial complex III; therefore, mitochondrial complexes I and III were the main protein complexes under selective pressure. To explore the function of these positive selection sites, all the positive selected amino acid sites are highlighted in Appendix A, showing the structure of the protein. Mitochondrial oxidative phosphorylation is an important life activity in organisms, providing energy for most cellular functions [75]. Mitochondrial Complex I (CI), the main entry point of nicotinamide adenine dinucleotide (NADH) electrons into the respiratory chain, is a large protein complex that is closely involved in energy metabolism [76]. In Complex I, the structural domain of Oxidored_q5_N was defined as NADH-ubiquinone oxidoreductase, which is the main entry point of electrons into the respiratory chain and closely related to energy metabolism [77]. Therefore, the mutation of ND series subunits could have a great impact on the transmission efficiency of mitochondria as well as on energy transfer and metabolic functions [78,79]. As for proton_antipo_M, containing Mnh1 and Mnh2, this family forms part of complex I, which is closely related to the translocation reaction of protons across the membrane and also affects energy transfer [80,81]. Mitochondrial Complex III (CIII) is a major multisubunit membrane-binding enzyme that is closely involved in ATP synthesis and respiratory energy transduction in many organisms [82]. Of the three catalytic subunits of CIII, only Cytb is a complete protein [83]. So *Cytb* mutations affect energy transduction in the cellular mitochondrial respiratory chain.

Due to the fact that positive selection only affects a small number of amino acid sites over a short evolutionary period, the signal of positive selection is frequently overwhelmed by successive negative selection at the majority of sites in a gene sequence, leading to the conclusion that there is no positive selection in the foreground branch when the branch model is analyzed [84]. However, the branch-site model is more likely to find that a foreground branch is subject to positive selection because it allows the amino acid site selection pressure to change [84,85]. Hence, the reason why positive selection sites were found only in the branch-site model in this study can also be explained by the above content.

Previous studies have also found that organisms exposed to low temperature stress may have engaged in positive selection of mitochondrial genes [86,87,88,89]. Sun et al. found that *Tetranychus truncatus* showed positive selection for ND4 in the process of adapting to low temperatures [90]. In addition, Xu et al. identified more positive selection sites under low temperature stress, using high-latitude Canadian Heptageniidae as foreground branches [30]. Similarly, our current study also recognized the influence of temperature on mitochondrial energy metabolism, but the influence was due more to cold adaptation in the past historical events of *Hyla* and *Dryophytes*. However, due to limited samples, the cause of this phenomenon still needs further analysis.

## 5. Conclusions

In this study, mitogenomes of *Hyla sanchiangensis* (except part of the D-loop) from two different sites were obtained. In addition, genetic distances and phylogenetic relationships between two different sites of *Hyla sanchiangensis* were also identified. This study also acknowledged the monophyly of *Hyla* and considered *Hyla* and *Dryophytes* to be sister groups. In the branch-site model, we found five positive selection sites with significant differences. According to the results, the cold adaptation experienced by *Hyla* and *Dryophytes* at historical events may be the cause of the positive selection of mitogenomes, possibly due to the need for more energy to adapt to low temperatures, but this conclusion needs to be confirmed by further research.

## Figures and Tables

**Figure 1 animals-13-01593-f001:**
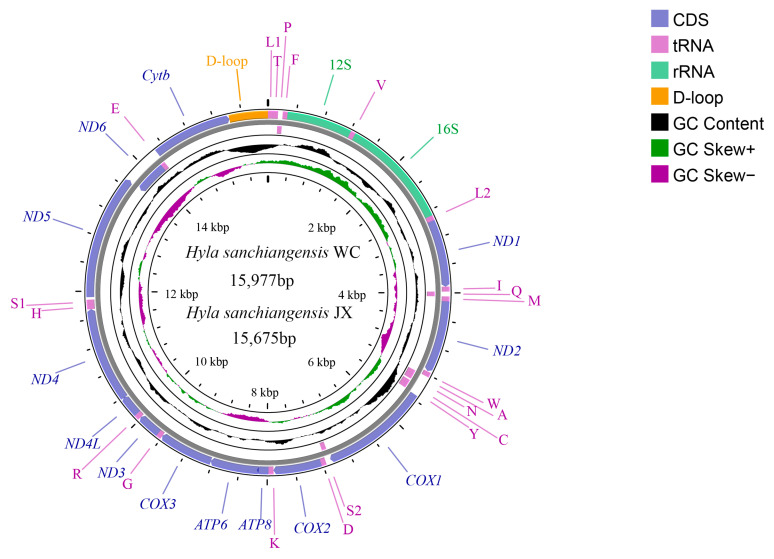
Mitogenome map of *H. sanchiangensis* JX and *H. sanchiangensis* LS. The outermost two circles show the gene map (PCGs, rRNAs, tRNAs, and control region) and genes; the outer circle is encoded by the positive strand, the second circle is encoded by the negative strand, and the tRNAs are all represented by abbreviations. The third circle represents GC content, and the green and violet parts of the innermost circle are GC skew. The GC content and GC skew are calculated using the deviation of the mean value of the whole series.

**Figure 2 animals-13-01593-f002:**
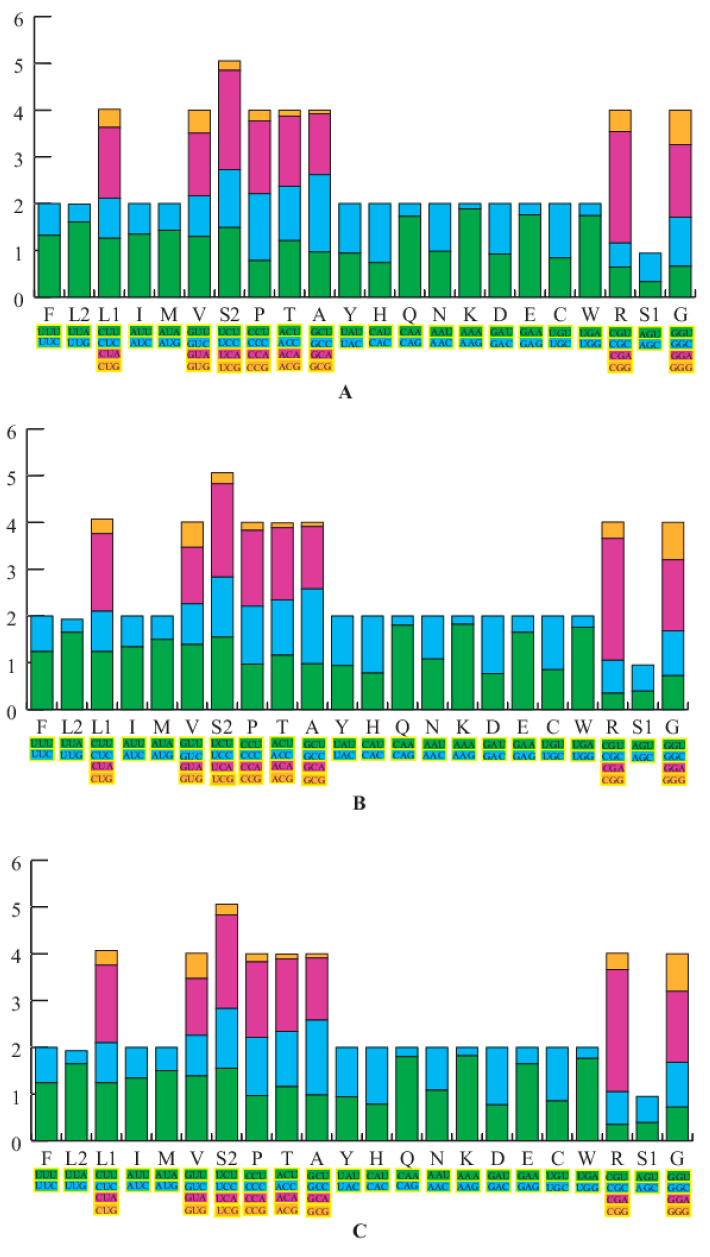
The relative synonymous codon usage (RSCU) in three *H. sanchiangensis* mitogenomes. The RSCU of the mitogenome is characterized for the individuals of *H. sanchiangensis* from Wencheng (**A**), Jinxiu (**B),** and Lishui (**C**). The X-axis shows all the codons used and different combinations of synonymous codons, with different codons represented by different colors, and the Y-axis lists the RSCU values.

**Figure 3 animals-13-01593-f003:**
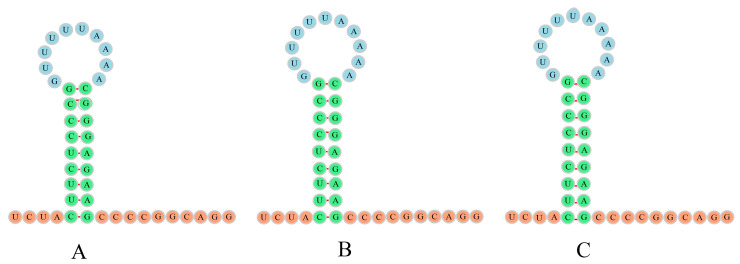
The secondary structures for L-strand origin of replication (O_L_) for the individuals of *H. sanchiangensis* from Jinxiu (**A**), Jinxiu (**B**), and Lishui (**C**).

**Figure 4 animals-13-01593-f004:**
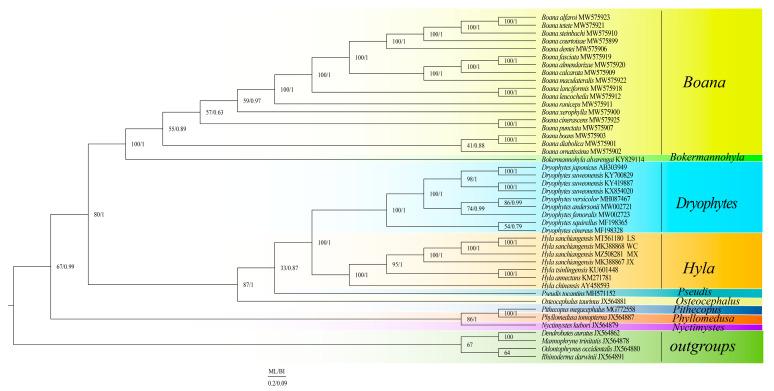
BI and ML analyses were used to predict the phylogenetic relationships of the Hylidae based on the nucleotide data set encoded by 13 proteins. Species name information and GenBank number are marked in the figure. Species of different genera are distinguished by different colors, and outgroups are represented by one color.

**Table 1 animals-13-01593-t001:** Genus names, GenBank accession numbers, and their references for the species used to construct the phylogenetic tree.

Family	Genus	Species	Length (bp)	GenBank Accession Number	Reference
Hylidae	*Bokermannohyla*	*Bokermannohyla alvarengai*	17,325	KY829114	[57]
	*Boana*	*Boana aff. courtoisae*	15,840	MW575899	[58]
		*Boana aff. lanciformis*	17,150	MW575918	
		*Boana alfaroi*	17,219	MW575923	
		*Boana almendarizae*	17,439	MW575920	
		*Boana boans*	17,021	MW575903	
		*Boana calcarata*	16,251	MW575909	
		*Boana cinerascens*	17,684	MW575925	
		*Boana dentei*	17,587	MW575906	
		*Boana diabolica*	14,606	MW575901	
		*Boana fasciata*	17,438	MW575919	
		*Boana leucocheila*	18,042	MW575912	
		*Boana maculateralis*	16,301	MW575922	
		*Boana ornatissima*	17,656	MW575902	
		*Boana punctata*	17,899	MW575907	
		*Boana raniceps*	17,180	MW575911	
		*Boana steinbachi*	18,107	MW575910	
		*Boana tetete*	16,107	MW575921	
		*Boana xerophylla*	18,062	MW575900	
	*Dendrobates*	*Dendrobates auratus*	14,963	JX564862	[32]
	*Dryophytes*	*Dryophytes andersonii isolate*	15,423	MW002721	[59]
		*Dryophytes cinereus voucher*	15,418	MF198328	[60]
		*Dryophytes femoralis isolate*	15,415	MW002723	[59]
		*Dryophytes suweonensis*	17,448	KX854020	[61]
			18,611	KY419887	[27]
			18,288	KY700829	Unpublished
		*Dryophytes versicolor*	18,800	MH087467	[26]
	*Hyla*	*Hyla annectans*	17,973	KM271781	[62]
		*Hyla chinensis*	18,180	AY458593	[63]
		*Hyla japonica*	19,519	AB303949	[64]
		*Hyla sanchiangensis* LS	15,664	MT561180	[31]
		*Hyla sanchiangensis* JX	15,675	MK388867	This study
		*Hyla sanchiangensis* WC	15,977	MK388868	This study
		*Hyla sanchiangensis* MX	15,694	MZ508281	Direct Submission
		*Hyla tsinlingensis*	18,305	KU601448	[65]
	*Nyctimystes*	*Nyctimystes kubori*	14,983	JX564879	[32]
	*Osteocephalus*	*Osteocephalus taurinus*	14,970	JX564881	[32]
	*Pithecopus*	*Pithecopus megacephalus*	18,050	MG772558	Unpublished
	*Phyllomedusa*	*Phyllomedusa tomopterna*	14,926	JX564887	[32]
	*Pseudis*	*Pseudis tocantins*	15,564	MH571152	[66]
Dendrobatidae	*Dendrobates*	*Dendrobates auratus*	14,863	JX564862	[32]
	*Mannophryne*	*Mannophryne trinitatis*	14,939	JX564878	
Alsodidae	*Odontophrynus*	*Odontophrynus occidentalis*	14,908	JX564880	
Rhinodermatidae	*Rhinoderma*	*Rhinoderma darwinii*	14,943	JX564891	

**Table 2 animals-13-01593-t002:** Locations of features in the mtDNA of *H. sanchiangensis* JX and *H. sanchiangensis* WC.

Gene/Region	Position JX/WC	Length (bp) JX/WC	Spacer (+) Overlap (−) JX/WC	Start Codon	Stop Codon	Strand
trnL1 (cta)	1–72	72	0			H
trnT (aca)	73–143	71	−1			H
trnP (cca)	143–211	69	−1			L
trnF (ttc)	211–278	68	0			H
12S rRNA	279–1211/279–1210	933/932	0			H
trnV (gta)	1212–1280/1211–1279	69	0			H
16S rRNA	1281–2884/1280–2876	1604/1597	0			H
trnL2 (tta)	2885–2957/2877–2949	73	0			H
*ND1*	2958–3918/2950–3910	945	0	TTG	T	H
trnI (atc)	3919–3989/3911–3981	71	−1			H
trnQ (caa)	3989–4059/3981–4051	71	−1			L
trnM (atg)	4059–4127/4051–4119	69	0			H
*ND2*	4128–5162/4120–5154	1029	+7	ATT	AGA	H
trnW (tga)	5170–5239/5162–5231	70	0			H
trnA (gca)	5240–5308/5232–5300	69	0			L
trnN (aac)	5309–5381/5301–5373	73				L
O_L_	5382–5406/5374–5399	27/28	0			
trnC (tgc)	5407–5470/5400–5463	64	0			L
trnY (tac)	5471–5540/5464–5533	70	+4			L
*COX1*	5545–7086/5538–7079	1527	+1	ATA	AGA	H
trnS2 (tca)	7088–7158/7081–7151	71	+1			L
trnD (gac)	7160–7228/7153–7221	69	+1			H
*COX2*	7230–7917/7223–7910	672	0	ATG	T	H
trnK (aaa)	7918–7990/7911–7982	73/72	0			H
*ATP8*	7991–8155/7983–8147	159	−25	ATG	TAA	H
*ATP6*	8131–8829/8123–8821	699	−1	ATC	TAA	H
*COX3*	8829–9614/8821–9606	783	−1	ATG	TAA	H
trnG (gga)	9614–9682/9606–9674	69	0			H
*ND3*	9683–10022/9675–10014	327	0	ATG	T	H
trnR (cga)	10023–10091/10015–10083	69	+2			H
*ND4L*	10094–10396/10086–10388	300	−7	ATG	TAG	H
*ND4*	10390–11754/10382–11746	1359	0	ATG	TAA	H
trnH (cac)	11755–11823/11747–11815	69	0			H
trnS1 (agc)	11824–11890/11816–11882	67	+36			H
*ND5*	11927–13729/11919–13721	1803	−17	ATG	AGA	H
*ND6*	13713–14210/13705–14202	495	0	ATG	AGA	L
trnE (gaa)	14211–14278/14203–14270	68	+2			L
*Cytb*	14281–15429/14273–15421	1149	0	ATG	TAG	H

**Table 3 animals-13-01593-t003:** The mitogenome composition of *H. sanchiangensis* from Wencheng, Jinxiu, and Lishui.

Region	*H. Sanchiangensis* WC				*H. sanchiangensis* JX				*H. sanchiangensis* LS (MT561180)			
	Length (bp)	AT%	AT-Skew	GC-Skew	Length (bp)	AT%	AT-Skew	GC-Skew	Length (bp)	AT%	AT-Skew	GC-Skew
Whole genome *	15,977	60.1	0	−0.275	15,675	59.9	0.002	−0.278	15,664	59.8	0.001	−0.275
PCGs	11,331	59.6	−0.075	−0.278	11,331	59.8	−0.074	−0.279	11,316	59.6	−0.078	−0.274
tRNA	1533	59.4	0.036	0.016	1533	59.9	0.053	−0.012	1528	59.3	0.047	−0.005
rRNA	2529	59.9	0.142	−0.120	2537	59.6	0.145	−0.125	2533	59.8	0.144	−0.119

Annotation: * means the whole genome except CR.

**Table 4 animals-13-01593-t004:** Adaptive analysis of the branch site model (BSM), in which *Hyla* and *Dryophytes* were used as foreground branches and other Hylidae species were used as background branches.

Model	np	LnL		Estimates of Parameters				Model Compared	LRT *p*-Value	Positive Sites
Model A	83	−169886.61	Site class	0	1	2a	2b	Model A vs. Model A null	0.000213429	1595 L 0.973 *, 2308 T 0.955 *, 2479 S 0.984 *, 2632 L 0.950 *, 3272 T 0.956 *
			f	0.9212	0.073	0.0056	0.00045			
			ω0	0.0427	1	0.043	1			
Model A null	82	−169893.47	ω1	0.0427	1	149.39	149.39			Not Allowed
			1							

Note: * indicates BEB > 0.95.

**Table 5 animals-13-01593-t005:** Characterization and description of positive selection sites of Hylidae species and comparison of amino acid sites in foreground and background branches.

Genes	Postive Selection Sites	Amino Acids		BEB Value	Feature Key	Description
		Foreground	Background			
Cytb	316	T/V/S	L	0.973 *	Domain	CYTB_CTER
ND3	85	V/A/S	T/A/S	0.955 *	/	/
ND4	47	N	S/P/A	0.984 *	Domain	Oxidored_q5_N
	200	T/M	L/V/M	0.950 *	Domain	Proton_antipo_M
ND5	400	V	T/A/S/V	0.956 *	Domain	Proton_antipo_M

Note: * indicates BEB > 0.95.

## Data Availability

Data to support this study are available from the National Center for Biotechnology Information (https://www.ncbi.nlm.nih.gov, accessed on 26 August 2022). The registration numbers are MK388867 and MK388868.

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
