# Peer review of "Characterization of Two Mitogenomes of Hyla sanchiangensis (Anura: Hylidae), with Phylogenetic Relationships and Selection Pressure Analyses of Hylidae"

_animals, 2023, doi:10.3390/ani13101593_

Round 1
Reviewer 1 Report
In this manuscript, the authors described two newly sequenced mitogenomes of the frog Hyla sanchiangensis from two distinct regions. They employed phylogenetic analyses to assess the relationships among the Hylidae group, making use of another 38 mitogenomes available in the NCBI database. Finally, the authors identified some evidences of positive selection operating in mitochondrial genes, which they attributed to cold stress adaptation.
I believe the authors address interesting questions in this work, the writing of the manuscript is smooth and I acknowledge its potential for publication. However, I identified some serious flaws that preclude me to recommend this manuscript for publication in its current state. Please, consider my following comments.
1) One of the key points in the present work consists in the assessment of the genetic differences among different populations (as stated in lines 27-29). The sequencing of mitogenomes is indeed a good approach, although it falls short due to the very limited sampling effort. This sampling is enough for phylogenetic analyses that aim to establish the relationships among different species, but for population studies the authors simply cannot assume that the single sampled individual is representative of the corresponding population, especially considering the highly genetic variability presented by mitochondrial genomes. The authors need an improved sampling for each population to assess the genetic distances within each population. Without it, the authors cannot assess the genetic distance between different populations.
2) Another goal of the authors is related to the clarification of the phylogenetic relationships of the Hylidae species.
2.1) In the Introduction is stated that the monophyly of the genera Hyla and Dryophytes lacks confirmation (lines 92-94). However, this monophyly was already shown in a previous study (please see Lima et al., 2017, doi: 10.1080/23802359.2017.1325339). A controversial point in Lima’s study was the clade composed of both Dryophytes immaculatus and Hyla tsinlingensis. In fact, the authors could had taken this opportunity to dispel this issue, but they did not include the corresponding H. tsinlingensis mitogenomes (NC_026524 and KP212702) in their analyses.
2.2) The authors also claimed that was Zhang et al. that suggested the subdivision of the Hyla taxon into the Hyla and Dryophytes genera (lines 95-96), but this suggestion was made originally by Duellman et al., 2016 (doi: 10.11646/zootaxa.4104.1.1).
2.3) It is not clear why the authors translated the sequences into amino acids after the alignment. It would be understandable if the authors translated the sequences before the alignment to maintain codon integrity or, alternatively, aligned the nucleotide sequences by codons.
2.4) Which test(s) of the DAMBE software did the authors used to assess nucleotide saturation? This should be explicit in the test and the results should be made available in the supplementary material.
2.5) Which analysis did the authors use to select GTR+I+G as the best-fit model? The RAxML software only employs a restricted list of nucleotide substitution models. For my perspective, a better option for a phylogenetic analysis software would be IQ-TREE, which includes the ModelFinder algorithm for the best-fit model assessment, and has the option for ultra-fast bootstraps.
3) The authors used branch-site models to detect positive selection operating on the branches of the Hyla + Dryophytes clade.
3.1) In lines 360-362, the authors claimed that the lack of positive selection can be inferred by the M0 model values and LRT p-values (0.0571 and <1, respectively). M0 is a reference model, and it is never used alone to infer the presence of positive selection. On the other hand, the LRTs are used to accept which of the nested models (the null or the alternative) represents the data in question, depending if the p-value is significant (<0.05) or non-significant (>0.05). I believe the authors mixed the concepts of p-value and ω-value, but nevertheless the M0 alone cannot identify events of positive selection. Also, which branch are the authors referring to with “this branch”?
3.2) Besides clade and branch-site models, the authors also used branch-models to detect positive selection. Contrarily to branch-site models, branch-models are very parameter-rich and their use is discouraged by the authors of PAML.
3.3) In lines 368-369, the position of the positively selected sites is given in relation to the total alignment. It would be better if the authors used the positions relative to the corresponding genes.
3.4) This is my biggest concern with this section of the manuscript. In the Discussion (lines 465-467), the authors attributed the positively selected sites to cold stress adaptation in Hyla and Dryophytes genera. Even though the authors adjusted their tone due to the lack of data, there is no evidence that the selected sites are, in any way, related to cold adaptation. Since site mutagenesis is not within the scope of this study, which would be the next logical step to test the authors’ hypothesis, the authors need to perform the same selection analyses on a control group (one with similar data size and genetic divergence to reduce unwanted bias). For example, in Zhou et al., 2014 (cited in this manuscript), the authors compared the mitogenomes of high-altitude and low-altitude birds. Without the proper control group, any claims made by the authors is just unfounded speculation, since the selected sites can be related with any evolutionary process independent from cold adaptation.
3.5) In lines 462-463, the authors misused a reference. Here, Cai et al. found negative selection in cold-acclimating species, while in the other references the authors found positive selection.
Minor remark:
- The statements in lines 61-63 need to be supported by suitable references.
Author Response
Reviewer #1:
- One of the key points in the present work consists in the assessment of the genetic differences among different populations (as stated in lines 27-29). The sequencing of mitogenomes is indeed a good approach, although it falls short due to the very limited sampling effort. This sampling is enough for phylogenetic analyses that aim to establish the relationships among different species, but for population studies the authors simply cannot assume that the single sampled individual is representative of the corresponding population, especially considering the highly genetic variability presented by mitochondrial genomes. The authors need an improved sampling for each population to assess the genetic distances within each population. Without it, the authors cannot assess the genetic distance between different populations.
Response: Thank you very much for your comments. In our previous study, we sequenced and aligned the COI gene of five to six Hyla sanchiangensis from these two populations, and we found that the genetic distance of these COI genes was 0-0.001% within the same site. So, we use one sample to obtain the complete mitochondrial genome of each population from two different sites. Based on the mt genome, we assessed the genetic distance of two mt genome. We want to use the two genetic distances of two mt genome to access the genetic distance of two populations. We use the “population” in the paper to represent that different site. In the future, we will also increase the number of samples to explore genetic differences between populations. Thank you for your comments again.
2) Another goal of the authors is related to the clarification of the phylogenetic relationships of the Hylidae species.
2.1) In the Introduction is stated that the monophyly of the genera Hyla and Dryophytes lacks confirmation (lines 92-94). However, this monophyly was already shown in a previous study (please see Lima et al., 2017, doi: 10.1080/23802359.2017.1325339). A controversial point in Lima’s study was the clade composed of both Dryophytes immaculatus and Hyla tsinlingensis. In fact, the authors could had taken this opportunity to dispel this issue, but they did not include the corresponding H. tsinlingensis mitogenomes (NC_026524 and KP212702) in their analyses.
Response: Thanks for your suggestion. In the past, Dryophytes was a subgenus of Hyla, Duellman et al., 2016 proposed that Dryophytes and Hyla were sister groups based on molecular evidence. However, this relationship lacks evidence other than molecular support, so we believe that further research is needed. In addition, after comparison, we believed that the sequence of NC_026524 and KP212702 were the same sequence with different GenBank Number, which was included in the initial tree formation (see Figure A) and the figure 1 in Lima et al., 2017. This is consistent with the results of these studies supporting Hyla suweoninsis and Hyla tsinlingensis (KU601448) as sister groups (e.g. Lee et al., 2017; Borzee et al., 2017; Nam et al., 2017; Lima et al., 2017), but this is all due to the use of KP212702. In Li et al., 2015 (doi.org/10.1016/j.ympev.2015.02.018), Yan et al., 2020 (doi.org/10.3389/fevo.2020.00234), Gatto et al., 2018, these studied use different Hyla tsinlingensis sequence, the results all showed that Hyla tsinlingensis and H. annectans are sister group relationships, which is the same as Hyla tsinlingensis (KU601448) in this study. In addition, the collection sites of Hyla tsinlingensis (KP212702 and KU601448) was Anhui Province. We also calculated the genetic distance between these two Hyla tsinlingensis (KU601448, KP212702) and found that the genetic distance was 22.3% (see Figure B). Therefore, we believe that the identification of KP212702 needs to be further studied, so it was not included in the final tree formation. And Zhang et al., 2019 (https://doi.org/10.1016/j.ijbiomac.2019.03.220) in the previous studies have proposed this idea. They thought the sequence of Hyla tsinlingensis (KP212702) is existed problem. We further blasted the 12S RNA gene sequence from typical collected sites of Hyla tsinlingensis (KP742645) with two Hyla tsinlingensis (KU601448, KP212702), and we found the sequence of Hyla tsinlingensis (KU601448) is right while the sequence of Hyla tsinlingensis (KP212702) is wrong. So, we did not use the wrong sequence to discuss the monophyly of Hyla tsinlingensis.
- The reconstructed phylogenetic relationship. But we think the the sequence of Hyla tsinlingensis (KP212702) is wrong.
- The distance of two mitogenomes of Hyla
2.2) The authors also claimed that was Zhang et al. that suggested the subdivision of the Hyla taxon into the Hyla and Dryophytes genera (lines 95-96), but this suggestion was made originally by Duellman et al., 2016 (doi: 10.11646/zootaxa.4104.1.1).
Reponse: Thank you very much for your comments. We have revised the sentences in the article according to your comments. The sentence here is ambiguous. We intended to express the point also raised by Zhang et al. (not the first time), and the part that caused the ambiguity has been corrected.
2.3) It is not clear why the authors translated the sequences into amino acids after the alignment. It would be understandable if the authors translated the sequences before the alignment to maintain codon integrity or, alternatively, aligned the nucleotide sequences by codons.
Response: Thank you very much for your advice. The main purpose of converting alignment sequences into amino acids is to find mutant amino acid sites in foreground and background branches.
2.4) Which test(s) of the DAMBE software did the authors used to assess nucleotide saturation? This should be explicit in the test and the results should be made available in the supplementary material.
Response: Thanks for your suggestion.We used Xia et al. 's method (https://doi.org/10.1093/jhered/92.4.371) to measure the substitution saturation (Nuc.seq.only), and detected the third codon saturation with a result of Iss<Iss.cSym,Iss<Iss.cAsym, and all the p < 0.05, which suggested that the third codon was not saturated and the results were credible. We put the result in the new Table S1.
2.5) Which analysis did the authors use to select GTR+I+G as the best-fit model? The RAxML software only employs a restricted list of nucleotide substitution models. For my perspective, a better option for a phylogenetic analysis software would be IQ-TREE, which includes the ModelFinder algorithm for the best-fit model assessment, and has the option for ultra-fast bootstraps.
Response: Thanks for your suggestion. In this study, we used Partitionfinder 1.1.1 software to select the optimal partitioning and best-fitting models for evolution, and the results of model selection are shown in Table S3.
3) The authors used branch-site models to detect positive selection operating on the branches of the Hyla + Dryophytes clade.
3.1) In lines 360-362, the authors claimed that the lack of positive selection can be inferred by the M0 model values and LRT p-values (0.0571 and <1, respectively). M0 is a reference model, and it is never used alone to infer the presence of positive selection. On the other hand, the LRTs are used to accept which of the nested models (the null or the alternative) represents the data in question, depending if the p-value is significant (<0.05) or non-significant (>0.05). I believe the authors mixed the concepts of p-value and ω-value, but nevertheless the M0 alone cannot identify events of positive selection. Also, which branch are the authors referring to with “this branch”?
Response: Thank you very much for your comments. We confused the concepts of p value and ω value. Branch mode (BM) did not receive a positive choice. After checking, we deleted this part.
3.2) Besides clade and branch-site models, the authors also used branch-models to detect positive selection. Contrarily to branch-site models, branch-models are very parameter-rich and their use is discouraged by the authors of PAML.
Response: Thank you very much for your comments. When we chose the Model at that time, we tended to choose more models for study, so the branch model was included. In addition, no positive selection sites were found in the Branch model in this study. We deleted that part.
3.3) In lines 368-369, the position of the positively selected sites is given in relation to the total alignment. It would be better if the authors used the positions relative to the corresponding genes.
Response: Thank you very much for your comments. We rewrote the position of the statement and made the loci correspond to the gene position for easy understanding.
3.4) This is my biggest concern with this section of the manuscript. In the Discussion (lines 465-467), the authors attributed the positively selected sites to cold stress adaptation in Hyla and Dryophytes genera. Even though the authors adjusted their tone due to the lack of data, there is no evidence that the selected sites are, in any way, related to cold adaptation. Since site mutagenesis is not within the scope of this study, which would be the next logical step to test the authors’ hypothesis, the authors need to perform the same selection analyses on a control group (one with similar data size and genetic divergence to reduce unwanted bias). For example, in Zhou et al., 2014 (cited in this manuscript), the authors compared the mitogenomes of high-altitude and low-altitude birds. Without the proper control group, any claims made by the authors is just unfounded speculation, since the selected sites can be related with any evolutionary process independent from cold adaptation.
Response: Thank you very much for your comments. This study mainly illustrates the results that Hyla and Dryophytes have positive selection sites for other species of Hylidae. In the discussion section, it is generally believed that temperature and oxygen (altitude) are the main factors affecting the selection pressure of animals. Such as Yu et al., 2011 (doi.org/10.1016/j.mito.2011.01.004), Li et al., 2018 (doi.org/10.3389/fgene.2018.00605). After comparison found in this study, the sequence to be used and no elevation height and other aspects of significant differences, the main difference on the latitude, so we think the temperature was the main factors affecting the mutation. However, whether it is caused by temperature needs to be confirmed by further experiments, so the writing in this part of the paper has been further changed. In addition, in our current studying, we used six species of Hylidae collected in China under the low temperature stress and detected their mitogenome expression levels by RT-PCR. We found that positive selected genes inferring to most ND genes showed a downward trend to six species of Hylidae under low temperature stress. In addition, the Hylidae under normal temperature and low temperature stress were compared, and the change was significant (p < 0.05) (we selected a Hylidae species to show its ND4 gene changes under low temperature stress and normal temperature conditions, as shown in Figure C), supporting the conclusion of this paper that the occurrence of positive selection sites is related to temperature.
3.5) In lines 462-463, the authors misused a reference. Here, Cai et al. found negative selection in cold-acclimating species, while in the other references the authors found positive selection.
Response: Thank you very much for your comments. This reference was deleted.
Minor remark:
- The statements in lines 61-63 need to be supported by suitable references.
Response: Thanks a lot for your suggestion. We added references here.

Reviewer 2 Report
The manuscript from Hong et al. sequenced two complete mitogenomes of Hyla sanchiangensis and reconstructed the phylogeny of Hylidae. The authors also detected positive selection in some sites of mitochondrial PCGs. The manuscript is well written and the results are clearly presented. Therefore I supported its publication but also recommend some minor revisions.
(1)Line 70, “A” should be lowercase.
(2)The method for ML analysis has already mentioned in lines 168-170, but the authors wrote again in lines 173-175. Please remain only one of them.
(3)Lines 343-435, the authors only sequenced one individual for each site (Lishui and Wencheng), and it's insufficient to conclude that the two populations had low genetic distances. Please rephrase the sentence.
(4)Lines 414-417, the authors supported the frogs into two clades Dryophytes and Hyla. Are there any morphological differences between the two clades? It's recommended to list a few other traits in addition to molecular data.
Author Response
Reviewer #2:
- Line 70, “A” should be lowercase.
Response: Thank you very much for your comments. We corrected the word in the article.
- The method for ML analysis has already mentioned in lines 168-170, but the authors wrote again in lines 173-175. Please remain only one of them.
Response: Thank you very much for your comments. We have deleted lines 173-175.
Lines 343-435, the authors only sequenced one individual for each site (Lishui and Wencheng), and it's insufficient to conclude that the two populations had low genetic distances. Please rephrase the sentence.
Response: Thank you very much for your comments. We have revised the sentences in the article according to your comments.
- Lines 414-417, the authors supported the frogs into two clades Dryophytes and Hyla. Are there any morphological differences between the two clades? It's recommended to list a few other traits in addition to molecular data.
Reponse: Thanks a lot for your suggestion. The genus Dryophytes and Hyla diverged from other Hylidae about 22.6 millions of years ago and this suggestion was made originally by Duellman et al., 2016 (doi: 10.11646/zootaxa.4104.1.1), but so far no definite morphological features were known to distinguish Hyla from Dryophytes.
Reviewer 3 Report
The manuscript is interesting and clearly written, plenty of different analyses had been performed. There are only small changes I can recommend.
93-94 “and full confirmation of the monophyly of the genus Hyla and genus Dryophytes using mitogenomes still needs full confirmation”.
It is written in the abstract that D-loop was also sequenced, but there are no words on D-loop in Materials and Methods. Only in Results section on line 203 is mentioned, that control region was partially sequenced.
167-170 “The GTR+I+G model from RAxML 8.2.0 [55] was selected for maximum likelihood (ML) analysis, with each node branch supported for evaluation under 1000 ultrafast replications.” and line 173 “RAxML 8.2.0 [55] was used for ML analysis, and the best-fit model of GTR+I+G was used” – repeated phrase, delete one.
228-233 Figure 1: I see, that third circle is black and white and the innermost circle is violet and green, but in the caption: “.The third circle represents GC content, and the black part of
the innermost circle is GC skew”, while according to the legend, GC skew is green and violet and GC content is black and white. Which one is right?
288-291 Figure 4: Although topologies were the same, please mention, which topology (ML or BI) was used as main one.
336-348 Please, provide more information on the results of mitogenomes assembling from RADseq data. Was it possible to assemble complete PCGs or there were gaps in them?
438 Please, change “niacinamide” to “nicotinamide”.
Author Response
Reviewer #3:
93-94 “and full confirmation of the monophyly of the genus Hyla and genus Dryophytes using mitogenomes still needs full confirmation”.
It is written in the abstract that D-loop was also sequenced, but there are no words on D-loop in Materials and Methods. Only in Results section on line 203 is mentioned, that control region was partially sequenced.
Response: Thanks a lot for your suggestion. The sequencing of D-loop in this experiment was similar to that of other PCGs, tRNAs and rRNAs, that is, universal primers were used for amplification, and then PrimerPrimer5.0 was used to design specific primers for amplification according to the obtained sequences. Due to the high repetition of D-loop, only part of the sequences was obtained by using specific primers in this study. The D-loop experimental methods and other parts are unified in the “materials and methods” part.
167-170 “The GTR+I+G model from RAxML 8.2.0 [55] was selected for maximum likelihood (ML) analysis, with each node branch supported for evaluation under 1000 ultrafast replications.” and line 173 “RAxML 8.2.0 [55] was used for ML analysis, and the best-fit model of GTR+I+G was used” – repeated phrase, delete one.
Response: Thank you very much for your comments. We have deleted one sentence in the article according to your comments.
228-233 Figure 1: I see, that third circle is black and white and the innermost circle is violet and green, but in the caption: “.The third circle represents GC content, and the black part of
the innermost circle is GC skew”, while according to the legend, GC skew is green and violet and GC content is black and white. Which one is right?
Reponse: Thanks a lot for your suggestion, GC skew is green and violet, we have changed it.
288-291 Figure 4: Although topologies were the same, please mention, which topology (ML or BI) was used as main one.
Response: Thanks a lot for your suggestion, we have added it.
336-348 Please, provide more information on the results of mitogenomes assembling from RADseq data. Was it possible to assemble complete PCGs or there were gaps in them?
Response: Thank you very much for your comments. The sequences used in this paper are in Table S4, while all RADseq data results have been provided in Table S6 and Table S7. In the process of splicing, some sequences that could not be spliced were left out. Finally, we obtained part of COI of the species of Hyla in Table S4. However, after comparison, it was found that these COI could be divided into two parts, so they were divided into two parts for genetic distance comparison when calculating genetic distance.
438 Please, change “niacinamide” to “nicotinamide”.
Response: Thanks for your suggestion, we have changed it.
Round 2
Reviewer 1 Report
I thank the authors for considering my previous comments. They addressed almost all point satisfactorily.
Still, I retain some concerns that are explained in the following paragraphs.
1) The authors assessed the genetic distance of two mitogenomes, sampled from different sites. There is nothing wrong with this, but the authors consistently use the term “population” which is misleading, especially in the abstract (line 28), transpiring that they used “population” data and analysed both sites from the “population” point of view. With just two samples, the authors are not able to assess the genetic distances of different populations. They need to revise the corresponding sections in order to be consistent with the data and results produced.
2) The authors still need to present a suitable control for the positive selection analyses, in order to support the hypothesis that they are due to cold stress adaptation. There is no evidence that another group of amphibians would not present a similar number of positive selection sites (or even more) in those mitochondrial genes. The authors’ argument that the differences observed in the expression levels support the hypothesis that the positive selection is a consequence of the cold stress adaptation is beyond my comprehension. How is the differential gene expression related to the biased amino acid mutations?
Author Response
Dear reviewers and editor:
Re: Manuscript ID: animals-2300965 and Title: Characterization of two mitogenomes of Hyla sanchiangensis (Anura: Hylidae), with phylogenetic relationships and selection pressure analyses of Hylidae. Thank you for your valuable comments on our articles. We read the comments carefully and corrected them. Thank you for giving us the opportunity to resubmit the revised manuscript.
Sincerly,
Dan-Na Yu and Yong-Pu Zhang
The responses to the reviewer’s comments are presented as following:
Reviewer #1:
I thank the authors for considering my previous comments. They addressed almost all point satisfactorily.
Still, I retain some concerns that are explained in the following paragraphs.
1) The authors assessed the genetic distance of two mitogenomes, sampled from different sites. There is nothing wrong with this, but the authors consistently use the term “population” which is misleading, especially in the abstract (line 28), transpiring that they used “population” data and analysed both sites from the “population” point of view. With just two samples, the authors are not able to assess the genetic distances of different populations. They need to revise the corresponding sections in order to be consistent with the data and results produced.
Response: Thank you for your suggestion. The part about "population" in the article all has been revised.
- The authors still need to present a suitable control for the positive selection analyses, in order to support the hypothesis that they are due to cold stress adaptation. There is no evidence that another group of amphibians would not present a similar number of positive selection sites (or even more) in those mitochondrial genes. The authors’ argument that the differences observed in the expression levels support the hypothesis that the positive selection is a consequence of the cold stress adaptation is beyond my comprehension. How is the differential gene expression related to the biased amino acid mutations?
Response: Thank you for your advice.
Since only one branch can be selected as the foreground branch in the branch-site model, only one branch can be selected even if there are multiple stressed branches to choose. For example, Zhou et al., 2014 (https://doi.org/10.1016/j.mito.2014.07.012) had four clades of high-altitude birds, but only one clade was taken as the foreground clade and the other clades were taken as the background during the selective stress analysis. The same situation also occurred in Li et al., 2018 (doi: 10.3389/fgene.2018.00605), and multiple clades were found in flying locusts. Therefore, we believe that although the selected sites in the background branch will weaken the selection in the foreground branch, the positive selection sites in the gene can still be represented.
We took each genus as the foreground Branch and the rest as the background branch. EasycodeML was used to conduct the selective pressure analysis of the branch-site model.
We found that positive selection sites were appeared when Bokermannohyla and Nyctimystes were used as foreground branches (show in the table below). The Hyla and Dryophytes' cold stress is due to their North American origin and spread to Eurasia through the Bering Strait, see Li et al., 2015 (https://doi.org/10.1016/j.ympev.2015.02.018). As for other species, Boana comes from the Amazon Basin, South America, including Venezuela, Guyana, French Guiana, Ecuador, Colombia, etc. (https://amphibiansoftheworld.amnh.org/). It was found that Boana was diversified throughout the Amazon during a historical event during the 10Ma period and did not experience cold stress (see Fouquet A et al., 2021 https://doi.org/10.1080/14772000.2021.1873869). Osteocephalus and Phyllomedusa live in the Amazon Basin of Ecuador, Brazil, Bolivia, Peru, and Colombia (https://amphibiansoftheworld.amnh.org/). It also occurs in the Amazon River basin and is thought not to experience cold stress (SCHIESARI L et al., 2022;DOI: 10.11646/ZOOTAXA.5223.1.1 ; Taucce P P G et al., 2022 https://doi.org/10.5852/ejt.2022.836.1919). Pseudis in Guianas, northeastern Venezuela (and expected in southeastern Venezuela), Trinidad, and southern Brazil, Paraguay, southeastern Peru, eastern Bolivia, northeastern Argentina, and Uruguay (https://amphibiansoftheworld.amnh.org/), apparently unrelated to cold stress. But in the study, the genetic differentiation of Pseudis tocantins was found to be related to rivers (see Fonseca E M et al., 2021 https://doi.org/10.1007/s10980-021-01257-z). Pithecopus, which lives in the east of the Andes from southern Venezuela to northern Argentina, is similarly immune to the effects of cold stress. Nyctimystes kubori widespread in the central highlands from approximately 141° to 147° E, as well as in the mountains of the Huon Peninsula, Papua New Guinea, 1100-2000 m elevation (https://amphibiansoftheworld.amnh.org/). So we think the positive selection site is mainly due to its altitude. In fact, the same thing happened in Li et al., 2018 (doi: 10.3389/fgene.2018.00605), where positive selection sites were also found in several flightless clades, which were later analyzed to be the result of other factors. Bokermannohyla alvarengai is also a high-altitude species, and it is affected by Quaternary climate fluctuations (specific studies can see https://doi.org/10.1016/j.ympev.2021.107113). Our detection results show that it has 7 positive selection sites, so cold stress in historical events will cause positive selection of the species. Of course, this species is still a high-altitude species, so there are more positive selection sites were found. We think this supports our study that cold stress from historical events leads to the production of positive selection sites. We also said in the previous section that due to the characteristics of branch-site model, only one foreground branch can be selected even if multiple branches are subjected to cold stress. Considering that Bokermannohyla alvarengai and Nyctimystes kubori are not the focus of this paper, their studies have also been pointed out in the previous literature (Francisco Fonseca Ribeiro de Oliveira et al., 2021 https://doi.org/10.1016/j.ympev.2021.107113), so they did not discuss in the text. As for the control group under cold stress, the current research data show that all species except Bokermannohyla alvarengai have not been subjected to cold stress duriing historical event (specific details can be seen before the description of each species).
So, when we used the clade of Hyla and Dryophytes as the foreground Branch (cold clade) and the rest as the background branch (none-cold clade), which can be considered as control clade. But we did not use this description in the paper because we also found the positive selection in the genera Bokermannohyla and Nyctimystes were used as foreground branches. It will make the paper some confuse.
Adaptive analysis of the Branch site model (BSM), in which Nyctimystes was used as foreground branches, and other Hylidae species were used as background branches.
|
Model |
np |
LnL |
Estimates of Parameters |
Model Compared |
LRT P-value |
Positive sites |
||||
|
Model A |
83 |
-169893.613015 |
Site class
|
0 |
1 |
2a |
2b |
Model A vs.Model A null |
0.011416241 |
58 S 0.746,251 L 0.703,1057 S 0.790,1882 T 0.507,2354 F 0.866,2458 V 0.610,2690 T 0.942,2741 V 0.655,2856 F 0.862,2860 I 0.688,2897 I 0.612,2923 L 0.955*,3234 L 0.537,3272 T 0.591,3410 T 0.835,3435 L 0.874,3584 M 0.822 |
|
f |
0.92177 |
0.073 |
0.0046 |
0.00037 |
||||||
|
ω0 |
0.04268 |
1 |
0.0426 |
1.00000 |
||||||
|
Model A null |
82 |
-169896.812688 |
ω1 |
0.04268 |
1 |
53.654 |
53.65366 |
Not Allowed |
||
|
1 |
|
|
|
|
||||||
Adaptive analysis of the Branch site model (BSM), in which Bokermannohyla was used as foreground branches, and other Hylidae species were used as background branches.
|
Model |
np |
LnL |
Estimates of Parameters |
Model Compared |
LRT P-value |
Positive sites |
||||
|
Model A |
83 |
-169834.698554 |
Site class
|
0 |
1 |
2a |
2b |
Model A vs.Model A null |
0.028070701 |
197 L 0.910,255 N 0.932,265 K 0.893,688 I 0.783,695 T 0.893,725 L 0.637,743 I 0.782,903 A 0.899,1015 S 0.898,1172 G 0.912,1242 K 0.957*,1248 H 0.948,1482 E 0.894,1493 S 0.961*,1513 A 0.763,1543 V 0.918,1636 L 0.801,1754 A 0.911,1791 P 0.921,1819 T 0.910,1824 I 0.909,1830 L 0.896,1877 N 0.784,1878 L 0.908,1884 L 0.910,1889 T 0.897,1901 L 0.586,1944 A 0.978*,1955 T 0.777,1975 S 0.904,1997 P 0.952*,2005 L 0.925,2081 L 0.932,2083 L 0.725,2104 N 0.926,2106 K 0.512,2128 T 0.916,2133 S 0.940,2138 L 0.894,2149 Q 0.963*,2178 T 0.783,2180 T 0.862,2185 L 0.933,2211 L 0.925,2226 T 0.787,2329 Q 0.953*,2465 L 0.839,2468 I 0.774,2525 V 0.784,2575 L 0.682,2609 V 0.917,2612 E 0.561,2734 L 0.665,2906 T 0.763,2931 S 0.980*,3059 L 0.912,3311 I 0.795,3317 T 0.898,3338 Q 0.924,3364 L 0.891,3499 S 0.915,3584 M 0.514 |
|
f |
0.88838 |
0.071 |
0.0378 |
0.00301 |
||||||
|
ω0 |
0.04186 |
1 |
0.0419 |
1.00000 |
||||||
|
Model A null |
82 |
-169837.110414 |
ω1 |
0.04186 |
1 |
2.0274 |
2.02740 |
Not Allowed |
||
|
1 |
|
|
|
|
||||||
